# Physicochemical Properties and In Vitro Digestibility of Starch from a Trace-Rutinosidase Variety of Tartary Buckwheat ‘Manten-Kirari’

**DOI:** 10.3390/molecules27196172

**Published:** 2022-09-20

**Authors:** Takahiro Noda, Koji Ishiguro, Tatsuro Suzuki, Toshikazu Morishita

**Affiliations:** 1Hokkaido Agricultural Research Center, NARO, Shinsei, Memuro, Kasai-gun, Hokkaido 082-0081, Japan; 2Kyushu-Okinawa Agricultural Research Center, NARO, Suya, Koshi, Kumamoto 861-1192, Japan

**Keywords:** Tartary buckwheat, rutinosidase, starch, physicochemical properties, digestibility

## Abstract

We recently developed a novel Tartary buckwheat variety, ‘Manten-Kirari’, with trace-rutinosidase activity. The use of ‘Manten-Kirari’ enabled us to make rutin-rich food products with low bitterness. This study was intended to evaluate the physicochemical properties and in vitro digestibility of starch isolated from ‘Manten-Kirari’. For comparison, the representative common buckwheat variety ‘Kitawasesoba’ and Tartary buckwheat variety ‘Hokkai T8’ in Japan were also used. The lowest content of amylose was found in ‘Manten-Kirari’ starch (18.1%) while the highest was in ‘Kitawasesoba’ starch (22.6%). ‘Manten-Kirari’ starch exhibited a larger median granule size (11.41 µm) and higher values of peak viscosity (286.8 RVU) and breakdown (115.2 RVU) than the others. The values of onset temperature for gelatinization were 60.5 °C for ‘Kitawasesoba’, 61.3 °C for ‘Manten-Kirari’, and 64.7 °C for ‘Hokkai T8’. ‘Manten-Kirari’ and ‘Hokkai T8’ starches were digested more slowly than ‘Kitawasesoba’ starch. Our results will provide fundamental information concerning the expanded use of ‘Manten-Kirari’ in functional foods.

## 1. Introduction

Buckwheat belongs to the Polygonaceae family and is grown as a traditional crop around the world. It has starch-rich grains that are processed, utilized, and consumed like cereals, for example, rice, wheat, and maize. Two popular buckwheat species, common buckwheat (*Fagopyrum esculentum* Moench) and Tartary buckwheat (*Fagopyrum tataricum* Gaertn.), are used for food and feed purposes. In Japan, buckwheat food products are mainly buckwheat noodles, which are made from a mixture of buckwheat wheat and flours. Buckwheat has high nutritional values of protein, dietary fiber, phenolic compounds, and minerals in its grains. Rutin, the major phenolic compound of buckwheat, is well known for its health-promoting properties, such as antioxidative [1,2], anti-hypertensive [3], and α-glucosidase inhibitory activities [4]. Tartary buckwheat has received extensive attention for its remarkably high rutin content as compared to that of common buckwheat [2,5]. However, Tartary buckwheat seeds exhibit markedly high rutinosidase activity, which easily hydrolyzes rutin to quercetin and rutinose within just a few minutes of adding water to Tartary buckwheat flour [6,7] (Figure 1). Products made from Tartary buckwheat contain several bitter compounds, leading to the limited use of Tartary buckwheat. The enzymatic hydrolysis of rutin plays a large role in the bitterness because quercetin, a rutin hydrolysate, is a bitter compound [8]. Recently, our research group released a new Tartary buckwheat variety, ‘Manten-Kirari’, that has trace-rutinosidase activity [9,10]. Our previous studies demonstrated that rutin-rich food products with low bitterness could be produced using ‘Manten-Kirari’ flour [11,12,13]. Additionally, Nishimura et al. [14] reported that the intake of rutin-rich noodles containing ‘Manten-Kirari’ lowered body weight, BMI (body mass index), and TBARS (2-thiobarbituric acid reactive substances) levels.

As starch is the major component of buckwheat grains, the starch properties have a great impact on the quality of buckwheat products. Comprehending the starch structure and functionality of buckwheat grains is critical for their suitable application in foods. The molecular structures and physicochemical properties of purified buckwheat starch have been studied extensively [15,16,17,18,19,20,21]. Review articles that deal with the present knowledge of buckwheat starch quality have been published in recent years [22,23]. However, no detailed studies of characteristics of starch from Tartary buckwheat grains with trace-rutinosidase activity have been carried out to date. Thus, in this study, the physicochemical properties and in vitro digestibility of starch from a trace-rutinosidase variety of Tartary buckwheat, ‘Manten-Kirari’, were evaluated and compared with those from a representative common buckwheat variety, ‘Kitawasesoba’, and a Tartary buckwheat variety, ‘Hokkai T8’, in Japan.

## 2. Results and Discussion

Table 1 presents data on the amylose content, median granule size, and color components of starches from two Tartary buckwheat varieties, ‘Manten-Kirari’ and ‘Hokkai T8’, and one common buckwheat variety, ‘Kitawasesoba’. The amylose content of ‘Manten-Kirari’ was manifestly lower (18.1%) than that of ‘Kitawasesoba’ (22.6%) and was slightly lower than that of ‘Hokkai T8’ (18.9%). These values were lower than those of previous research on buckwheat starch [15,17,19,20,21]. This could be due to differences in variety, cultivation conditions, and/or methods for measuring amylose content. Generally, the amylose content affects the physicochemical parameters of a starch because amylose possesses the ability to form a firm gel. The median granule size differed significantly among buckwheat varieties. ‘Manten-Kirari’ starch had the largest median granule size (11.41 µm), whereas ‘Kitawasesoba’ starch had the smallest (8.14 µm). In our present study, the median granule sizes of buckwheat starches were somewhat larger than those reported by Zheng et al. [18] and Qian et al. [19], who revealed that the average granule size of common buckwheat starch is around 6 µm. Values of L* (lightness), a* (greenness–redness), and b* (blueness–yellowness) are given as evaluated by a color meter. All buckwheat starches examined exhibited higher L* (89.87–91.48) as well as low a* (0.70–1.18) and b* (1.78–3.79). Higher L* values (close to 100) represented strong white color, while a* and b* values were close to zero, which indicated a neutral color. Thus, it was suggested that all of the starches had satisfactorily high purity. ‘Manten-Kirari’ starch showed a lower L* value (89.87) than that of ‘Hokkai T8’ (91.32) and ‘Kitawasesoba’ (91.48) starches. Higher b* values were observed in ‘Manten-Kirari’ (3.79) and ‘Hokkai T8’ (3.29) starches than in ‘Kitawasesoba’ (1.78), indicating that two Tartary buckwheat starches had a slightly stronger yellow cast. Li et al. [15] studied the color parameters of common and Tartary buckwheat starches, and they observed that Tartary buckwheat starches were more yellow than common buckwheat starches, as shown by our data. This suggested that the yellowness could not be removed completely during the isolation of starch from Tartary buckwheat flour.

Rapid visco-analyzer (RVA), in which viscosity is monitored over time during a standard heating–cooling cycle under a constant rate of shear, is becoming popular for analyzing the pasting properties of starches. The pasting properties of buckwheat starches assessed by RVA with 8% starch suspension (dry weight basis, *w*/*w*) are given in Table 2. The peak viscosity, breakdown, setback, and pasting temperature differed significantly among the buckwheat starches examined. ‘Manten-Kirari’ starch displayed the highest values of peak viscosity (286.8 RVU) and breakdown (115.2 RVU), while ‘Kitawasesoba’ starch had the lowest (peak viscosity, 251.1 RVU; breakdown, 80.4 RVU). This implied that ‘Manten-Kirari’ starch had greater swelling during heating and lower resistance to shearing at a high temperature. The setback, the ability of starch granules to re-associate after heating and cooling, varied in the following order: ‘Kitawasesoba’ (128.4 RVU) > ‘Manten-Kirari’ (119.0 RVU) > ‘Hokkai T8’ (109.6 RVU). The lowest pasting temperature was observed in ‘Manten-Kirari’ (71.9 °C) starch, while the highest was in ‘Hokkai T8’ starch (73.8 °C). Starch-pasting properties have been shown to be influenced by the amylose content, as starch swelling is inhibited by amylose. Generally, higher values of peak viscosity and breakdown and reduced values of setback and pasting temperature are associated with lower amylose content [24,25]. Our present data on peak viscosity, breakdown, and pasting temperature supported this concept. However, the setback of ‘Manten-Kirari’ starch with the lowest amylose content was not the highest.

Differential scanning calorimetry (DSC), which determines thermal properties, has been used to analyze the gelatinization properties of starches. The gelatinization properties of buckwheat starches monitored by DSC with 30% starch suspension (dry weight basis, *w*/*w*) are presented in Table 3. ‘Manten-Kirari’ starch showed lower onset temperature (To) (61.3 °C) and peak temperature (Tp) (67.1 °C) for gelatinization than ‘Hokkai T8’ starch (To, 64.7 °C; Tp, 69.3 °C), whereas ‘Kitawasesoba’ starch exhibited lower To (60.5 °C) and Tp (66.4 °C) than ‘Manten-Kirari’ starch. No significant difference in enthalpy (∆H) (10.6–11.1 J/g) for gelatinization was recognized among buckwheat starch samples examined. Our present results of To and Tp were comparable to the values reported previously [15,18,19,20,21]. Li et al. [15] and Qian et al. [19] obtained the results that buckwheat starches had ∆H of around 10 J/g, which were in good agreement with our present data. Zhang et al. [18] and Yoshimoto et al. [20] observed higher ∆H of 12.7 J/g and 14.5–15.0 J/g in buckwheat starches, respectively, while lower ∆H of 7.5–8.3 J/g was reported by Gao et al. [20]. Contrary to these results, we previously found great diversities in To (51.5–62.3 °C), Tp (57.2–66.7 °C), and ∆H (9.4–13.9 J/g) among starches from 17 samples of common buckwheat and 10 samples of Tartary buckwheat [17]. The differences in DSC gelatinization parameters between the present and previous data on buckwheat starches might be attributed to differences in variety, cultivation conditions, and/or experimental conditions for DSC analysis. The decrease in the amylose content, which indicates a high relative crystal concentration, usually causes higher ∆H [25]. In contrast, ‘Manten-Kirari’ starch with the lowest amylose content did not show higher ∆H.

Starch resistant to amylolytic enzymes has been associated with a decrease in postprandial glycemic responses, resulting in reductions in obesity, diabetes, and cardiovascular disease. Previous studies found that buckwheat products had lower rates of starch hydrolysis as compared with those of wheat products [26,27]. Therefore, buckwheat shows promise as a material of functional food based on its low starch digestibility. However, little information is available on the in vitro digestibility of the purified buckwheat starch granules except for the report of Acquistucci and Fornal [16]. Thus, we performed in vitro enzymatic digestion of the purified buckwheat starch granules by porcine pancreatic amylase and amyloglucosidase at different time points (20, 60, 120, and 240 min), and the results are shown in Figure 2. A rapid enhancement of the hydrolysis percentage of three buckwheat starches was found during the first hydrolysis time of 20 min. The hydrolysis percentage of three buckwheat starches increased gradually with hydrolysis times of 20–240 min. Namely, hydrolysis percentages at 20, 60, 120, and 240 min were observed to be in the ranges of 25.3–27.2%, 49.1–55.7%, 76.5–81.4%, and 95.5–98.3%, respectively. ‘Manten-Kirari’ and ‘Hokkai T8’ starches exhibited lower hydrolysis patterns than that of ‘Kitawasesoba’ starch. Until a hydrolysis time of 120 min, ‘Manten-Kirari’ starch showed a hydrolysis pattern similar to that of ‘Hokkai T8’ starch. However, for the hydrolysis time of 240 min, the hydrolysis percentage of ‘Manten-Kirari’ starch was somewhat lower (95.5%) than that of ‘Hokkai T8’ starch (97.1%). Assumedly, ‘Manten-Kirari’ containing starch slightly resistant to amylolytic enzymes at a late hydrolysis stage could be beneficial for health. It is well known that granule size and amylose content can affect the enzymatic digestibility of native starch granules. Smaller starch granules have been reported to digest faster than large granules [28,29,30,31]. Starch granules with lower amylose content have been shown to have higher enzymatic digestibility [16,25,32,33,34]. Thus, the higher digestibility of ‘Kitawasesoba’ starch could be related to its smaller granule size. However, contradictory data—that ‘Manten-Kirari’ starch with a lower amylose content did not display higher digestibility—were obtained in this study. Our present results of in vitro enzymatic digestion suggested that granule size had a stronger effect on starch digestibility than did the amylose content.

Most domestic buckwheat in Japan is marketed as flour and is used to manufacture noodles called soba. Previously, we have developed rutin-rich noodles from a novel Tartary buckwheat variety, ‘Manten-Kirari’, with trace-rutinosidase activity [12,13]. Starch properties appear to be the most critical determinant of the quality of buckwheat noodles, as starch is the main component. The present study has focused on the characteristics of starch prepared from ‘Manten-Kirari’ for its expanded use in rutin-rich foods, especially noodles. It is confirmed that ‘Manten-Kirari’ starch has lower amylose content, larger granule size, higher peak viscosity, and relatively lower enzymatic digestibility as compared with other varieties. Reduced amylose content [35,36,37] and increased peak viscosity [36,38] of starch are beneficial to the quality of white salted noodles made from wheat flour. ‘Manten-Kirari’, containing starch with low amylose content and high peak viscosity, would be desirable for making rutin-rich buckwheat noodles with good texture. As the relationship of starch properties with the quality of buckwheat noodles remains unknown, further investigation would be needed.

## 3. Materials and Methods

### 3.1. Materials

Tartary buckwheat flours from the varieties ‘Manten-Kirari’ and ‘Hokkai T8’ were purchased from Kobayashi Shokuhin Co., Ltd., Okoppe, Hokkaido, Japan. The common buckwheat variety ‘Kitawasesoba’ was grown in an experimental field at the Hokkaido Agricultural Research Center, NARO, Memuro, Hokkaido, Japan. Common buckwheat flour from ‘Kitawasesoba’ was obtained by milling the grains. Starch was isolated from each buckwheat flour using 0.2% sodium hydroxide as described previously [17].

### 3.2. Physicochemical Properties of Starch

The amylose content was estimated based on the blue value at 680 nm in accordance with the method previously described [17]. The median granule size was determined using a Sympatec HELOS Particle Size Analyzer by a previously reported method [39]. Color value analysis was carried out using a color meter as described previously [40]. Pasting properties and gelatinization properties were measured using RVA and DSC, respectively, as described earlier [41], except that RVA measurement was performed with 8% starch suspension (dry weight basis, *w*/*w*), not with 4%.

### 3.3. In Vitro Digestibility of Starch

In vitro digestibility of starch was determined by the Megazyme Resistant Starch Assay Kit, 05/2008 (Megazyme International Ireland Ltd., Co. County Wicklow, Ireland) in accordance with AOAC method 2002.02 [42]. After digestion at given time points (20, 60, 120, and 240 min), the percentage of starch digestion was calculated by measuring the content of glucose in the digestion solution.

### 3.4. Statistical Analysis

The determinations of all starch quality parameters were conducted in triplicate. Duncan t-tests were computed to measure variations in the average of all starch quality parameters of each variety.

## 4. Conclusions

The physicochemical properties and in vitro digestibility of starches isolated from a novel Tartary buckwheat variety, ‘Manten-Kirari’, with trace-rutinosidase activity, as well as the common buckwheat variety, ‘Kitawasesoba’, and a Tartary buckwheat variety, ‘Hokkai T8’, were determined. ‘Manten-Kirari’ starch exhibited the lowest amylose content, largest median granule size, and highest peak viscosity. The hydrolysis rates of ‘Manten-Kirari’ and ‘Hokkai T8’ starches were lower than that of ‘Kitawasesoba’ starch. Some properties, such as lower amylose content and higher peak viscosity, of ‘Manten-Kirari’ starch may be advantageous for making buckwheat noodles with good texture. Moreover, the beneficial health effects of ‘Manten-Kirari’-containing foods derived from the content of starch fraction resistant to amylolytic enzymes as well as the rutin content can be expected.

## Figures and Tables

**Figure 1 molecules-27-06172-f001:**
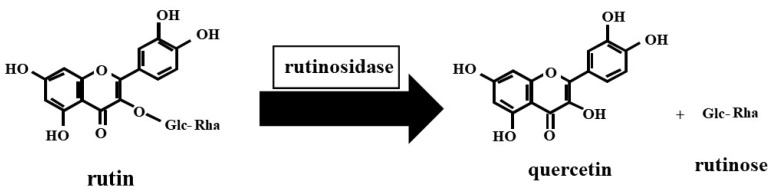
Rutin hydrolysis in Tartary buckwheat flour.

**Figure 2 molecules-27-06172-f002:**
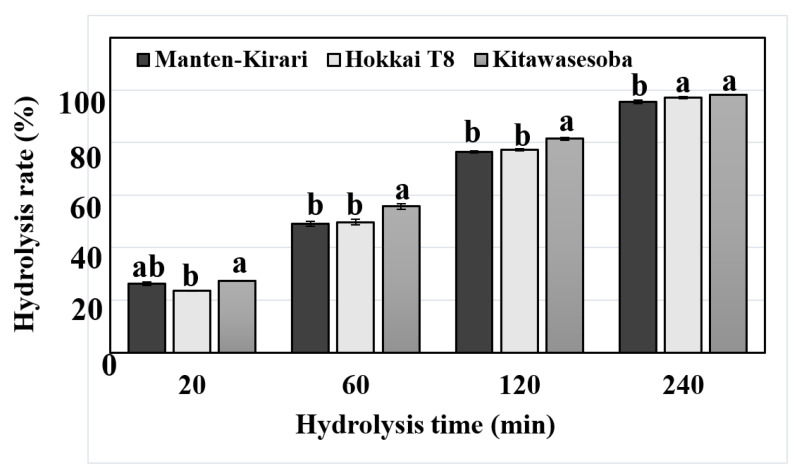
Hydrolysis rate of buckwheat starches by amylases. Values are means (n = 3). Error bars indicate SD. For each hydrolysis time, values with similar letters do not differ significantly (*p* > 0.05).

**Table 1 molecules-27-06172-t001:** Amylose content, median granule size, and color components (L*, a*, and b*) of buckwheat starches.

			Color Components		
Varieties	Amylose Content (%)	Median Granule Size (µm)	L*	a*	b*
Manten-Kirari	18.1 ± 0.2 b	11.41 ± 0.02 a	89.87 ± 0.09 b	1.18 ± 0.04 a	3.79 ± 0.11 a
Hokkai T8	18.9 ± 0.7 b	9.93 ± 0.01 b	91.32 ± 0.07 a	1.09 ± 0.02 a	3.29 ± 0.05 b
Kitawasesoba	22.6 ± 1.4 a	8.14 ± 0.00 c	91.48 ± 0.05 a	0.70 ± 0.00 b	1.78 ± 0.04 c

Values are means ± SD (n = 3). Means with similar letters in a column do not differ significantly (*p* > 0.05).

**Table 2 molecules-27-06172-t002:** RVA pasting properties of buckwheat starches.

Varieties	Peak Viscosity (RVU)	Breakdown (RVU)	Setback (RVU)	Pasting Temperature (°C)
Manten-Kirari	286.8 ± 1.1 a	115.2 ± 4.1 a	119.0 ± 0.4 ab	71.9 ± 0.0 b
Hokkai T8	274.9 ± 1.4 b	104.3 ± 1.2 b	128.4 ± 4.7 a	73.8 ± 0.5 a
Kitawasesoba	251.1 ± 1.4 c	80.4 ± 1.9 c	109.6 ± 2.3 b	72.6 ± 0.1 b

Values are means ± SD (n = 3). Means with similar letters in a column do not differ significantly (*p* > 0.05).

**Table 3 molecules-27-06172-t003:** DSC gelatinization properties of buckwheat starches.

Varieties	To (°C)	Tp (°C)	ΔH (J/g)
Manten-Kirari	61.3 ± 0.2 b	67.1 ± 0.4 b	10.6 ± 0.3 a
Hokkai T8	64.7 ± 0.1 a	69.3 ± 0.1 a	11.0 ± 0.3 a
Kitawasesoba	60.5 ± 0.3 c	66.4 ± 0.2 b	11.1 ± 0.2 a

Values are means ± SD (n = 3). Means with similar letters in a column do not differ significantly (*p* > 0.05).

## Data Availability

Not applicable.

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
