# Peer review of "Physicochemical Properties and In Vitro Digestibility of Starch from a Trace-Rutinosidase Variety of Tartary Buckwheat ‘Manten-Kirari’"

_molecules, 2022, doi:10.3390/molecules27196172_

Round 1

Reviewer 1 Report

The physicochemical properties and in vitro digestibility of starch from a Trace-rutinosidase Variety of Tartary Buckwheat ‘Manten-Kirari’ were investigated. The research is interesting and practical. But before it can be accepted by Molecules, it needs some revisions.:

1) Basic information about these buckwheat varieties such as starch content, protein content, fat content, etc. should be provided.

2) For the food application, not only the digestibility and pasting properties of starch are important, but also its functional properties such as paste clarity, freeze-thaw stability. If possible, add relevant content.

Author Response

Dear Sir/Madam

Here I am sending.

Best regards

Takahiro Noda

Reviewer 2 Report

In the described study the authors investigated the physicochemical properties and in vitro digestibility of starches isolated from a novel Tartary buckwheat variety, ‘Manten-Kirari’, with trace-rutinosidase activity, and compared it with the common buckwheat variety, ‘Kitawasesoba’, and a Tartary buckwheat variety‘Hokkai T8’. They found that starch from ‘Manten-Kirari’ variety had the lowest amylose content, largest median granule size, and highest peak viscosity. Also, they found that the hydrolysis rates of ‘Manten-Kirari’ and ‘Hokkai T8’ starches were lower than that of ‘Kitawasesoba’ starch. The authors propose that ‘Manten-Kirari’ starch with these properties, such as lower amylose content and higher peak viscosity, could be favorable for preparing buckwheat noodles with good texture. Moreover, the authors expect the beneficial health effects of ‘Manten-Kirari’-derived foods caused by the content of starch fraction resistant to amylolytic enzymes and coupled with the rutin content.

 This work is relevant to the field because, no detailed studies of characteristics of starch from Tartary buckwheat grains with trace-rutinosidase activity have been carried out before and the authors performed a study to investigate this and to try to find starches from novel buckwheat grains with good cooking properties and with beneficial health effects.

 The experiments in the study were well planned and executed and the resulting data is presented in a good and clear manner.

The drawn conclusions are appropriate and supported by the data presented in the paper.

The cited references are relevant and sufficient.

 The manuscript is well written and easy to read and does not need any corrections.

Author Response

(The authors gave the same response as above.)

Reviewer 3 Report

The paper entitled “Physicochemical properties and in vitro digestibility of starch from a trace-rutinosidase variety of Tartary buckwheat ‘Manten-Kirari’” revealed the physicohemical properties of ‘Manten-Kirari,’ such as the lowest amylose content, the largest granule size, and the highest peak viscosity in the tested samples. This paper also showed the starch from ‘Manten-Kirari’was slightly resistant against the enzymatic hydrolysis and discussed the correlation of the physicochemical properties and hydrlosysis rate. This study can attract the interest of many food scientists, not only readers of the journal molecules. I recommend this article for publication at the current form.

Author Response

(The authors gave the same response as above.)

Round 2

Reviewer 1 Report

The authors have carefully revised this manuscript. I think that it can be accepted.